# Simplifying SARS-CoV-2 wastewater-based surveillance using an automated FDA EUA assay

Shivaprasad H. Sathyanarayana,[1] Ashlee A. Robins,[1] Diana M. Toledo,[2] Torrey L. Gallagher,[1] Gregory J. Tsongalis,[1] Jacqueline A. Hubbard,[3] Joel A. Lefferts,[1] Isabella W. Martin[1]

**ABSTRACT** Wastewater-based surveillance (WBS) can track the spread of severe acute respiratory syndrome coronavirus 2 (SARS-CoV-2) in communities. Laboratory methods for this testing involve labor-intensive, multi-step processes. This study assessed the feasibility of performing WBS with an off-label use of an automated commercial SARS-CoV-2 assay that had received Emergency Use Authorization for human diagnostic testing from the United States Food and Drug Administration (FDA EUA). Twenty-four-hour composite samples of primary influent wastewater from seven municipalities in New Hampshire and Vermont were collected between September 2020 and February 2021, and were centrifuged upon receipt. An aliquot of fresh supernatant was immediately tested with the Abbott *m*2000 RealTi*m*e SARS-CoV-2 assay (Abbott Molecular, Des Plaines, IL, USA). Corresponding aliquots were then stored at −80°C until they were thawed, polyethylene glycol (PEG) concentrated, and tested by two PCR-based laboratory-developed tests (LDTs). Wastewater samples (103) were tested with successful detection of SARS-CoV-2 viral RNA by all three methods. Bland-Altman analysis showed overall concordant results with a bias of −0.13 and −0.42 log copies/mL detected by the FDA EUA assay compared to the LDTs. Specimen stability assessment demonstrated a decrease of 33.9% measurable viral RNA after three freeze-thaw cycles. SARS-CoV-2 detection in wastewater using an FDA EUA assay on an automated commercial testing platform performed comparably but with more efficient workflow when compared to two LDTs. This sample-to-answer automated method could save time and labor for surveillance testing, but further validation of its ability to quantitate SARS-CoV-2 viral RNA is necessary.

**IMPORTANCE** This proof-of-principle study evaluates an off-label use of an automated United States Food and Drug Administration (FDA) Emergency Use Authorization (EUA) severe acute respiratory syndrome coronavirus 2 (SARS-CoV-2) human diagnostic assay for wastewater surveillance. Compared to standard, labor-intensive, multi-step methods currently in use for wastewater surveillance testing, an off-label use of an FDA EUA assay on an automated platform offers a sample-to-answer testing requiring less labor and a faster turnaround time.

**KEYWORDS** SARS-CoV-2, COVID-19, wastewater, laboratory automation

Wastewater-based surveillance (WBS) became a public health tool to track the spread of coronavirus disease 2019 (COVID-19) in communities (1, 2) because infected individuals who are symptomatic or asymptomatic can shed the severe acute respiratory syndrome coronavirus 2 (SARS-CoV-2) virus in their feces (3, 4). In the USA, the National Wastewater Surveillance System was launched in September 2020 to build national capacity to test wastewater treatment plant (WWTP) influent for early detection of viral spread and emerging variants of concern in communities (5).

**Peer Reviewer** Megan Elizabeth Jane Lott, University of Georgia, Athens, Georgia, USA

Address correspondence to Isabella W. Martin, Isabella.W.Martin@hitchcock.org.

The authors declare no conflict of interest.

During the global emergence of SARS-CoV-2 in early 2020, there was an urgent need for rapid development of standardized laboratory methods for both human diagnostic testing and WBS. The overwhelmingly high volume of clinical samples led to rapid adaptation and scaling of sample-to-answer automated testing platforms from multiple commercial manufacturers for the purpose of human diagnostic testing (6). In contrast, WBS continues to rely on laboratory-developed tests (LDTs), which involve labor-intensive, multi-step processes. WBS LDTs typically start with one or more pre-processing steps, then use either reverse transcriptase real-time polymerase chain reaction (RT-qPCR) or reverse transcriptase droplet digital PCR (RT-ddPCR) as benchmark techniques for detecting and quantifying SARS-CoV-2 RNA in wastewater samples (2, 7, 8). Several factors can result in quantification bias when performing this testing in wastewater, including sample quality, sample transport, storage conditions, anticipated loss of viral RNA during pre-processing steps, and variability in protocols, instrumentation, reagents, data analysis, and interpretation (8, 9). Moreover, specific PCR inhibitors such as pharmaceutical agents, personal care products, household detergents, and industrial effluents can affect the efficiency of target amplification, detection, and quantification in wastewater (10–12).

In March 2020, Abbott Molecular (Des Plaines, IL, USA) was one of several manufacturers to develop and receive Emergency Use Authorization (EUA) from the United States Food and Drug Administration (FDA) for an assay that automates both sample preparation and RT-qPCR for qualitative detection of SARS-CoV-2 viral RNA in nasopharyngeal and oropharyngeal swabs from individuals with suspected COVID-19 infection (13, 14). The assay amplifies target regions of the SARS-CoV-2 RNA-dependent RNA polymerase (*RdRp*) gene and the *N* gene with the same fluorophore within a single well (14). It is run on an automated, sample-to-answer platform: the *m*2000 Realti*me* PCR System. Up to 96 samples can be processed simultaneously, with 470 test results reported in 24 hours (14). While the EUA of this assay is limited to specific specimen types, off-label validation and testing of alternative liquid specimens can be performed. We refer to an off-label use of this assay for WBS as the "FDA EUA assay" throughout the paper.

In this evaluation study, we compared three laboratory workflows for detecting and quantifying SARS-CoV-2 RNA in wastewater samples from urban and rural municipalities of New Hampshire and Vermont: (i) the FDA EUA assay described above, (ii) an RT-qPCR LDT, and (iii) an RT-ddPCR LDT. In addition, we also evaluated sample stability in different storage conditions for the FDA EUA assay.

## MATERIALS AND METHODS

### Collection of wastewater samples and sites

Twenty-four-hour composite wastewater influent samples were collected twice weekly from September 2020 through February 2021 from WWTP in the New Hampshire municipalities of Lebanon, Hanover, Concord, Nashua, and Woodsville, and the Vermont municipalities of Hartford and Burlington (three municipal WWTPs). Once collected, each 200mL sample was transported on the same day it was collected to our laboratory for processing, testing, and analysis.

### Processing wastewater samples

All wastewater samples were processed as previously described (15). In brief, samples were centrifuged for 30 minutes at 4,000 × *g* at 4°C, and the "clarified" supernatant was transferred into multiple aliquot tubes: one was used immediately for the FDA EUA assay with no further pre-processing steps, and the rest were frozen at −80°C until further testing by the LDTs (Fig. 1).

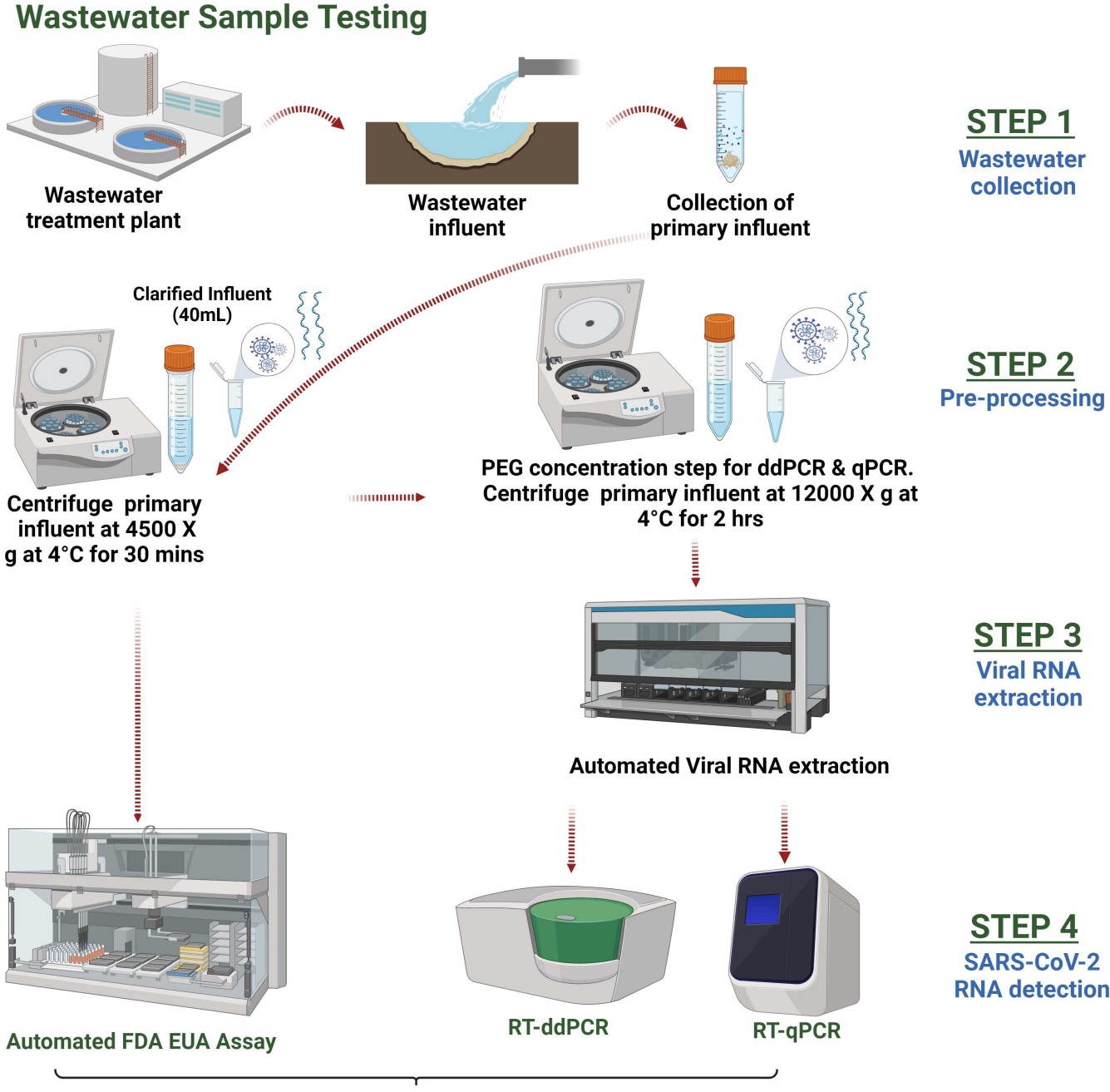

**FIG 1** Illustration of workflow for sample collection, pre-processing, viral RNA extraction, and SARS-CoV-2 detection using the FDA EUA assay, RT-ddPCR, and RT-qPCR methods.

## SARS-CoV-2 testing with the FDA EUA assay

A 2 mL aliquot of each clarified wastewater supernatant was tested on the same day as specimen receipt using the FDA EUA assay. All sample processing and runs were carried out according to the manufacturer's standard protocol with negative and positive external controls tested with each run to ensure acceptable assay performance. In this assay, RNA extraction, preparation, and RT-qPCR detection occur as part of the *m*2000 RealTi*me* System (Abbott). The assay adds an RNA sequence unrelated to the SARS-CoV-2 sequence to each sample that is amplified and serves as an internal control to confirm

proper RNA extraction and target amplification and detection. Although the FDA EUA assay is marketed as a qualitative test, cycle threshold (Cq) values are available to the user. We employed SeraCare quantitative calibration material and positive pooled sample dilutions to create a dilution series and calibration curve that was tested against reference material spiked into negative clarified wastewater samples (Supplemental Methods, Tables S1 and S2; Fig. S1 and S2).

## Wastewater sample concentration and SARS-CoV-2 viral RNA extraction for LDTs

After thawing, aliquots of the clarified wastewater supernatant underwent polyethylene glycol 8000 (PEG) concentration and extraction (15, 16). In brief, PEG and NaCl were added for a final concentration of 10% and 2.25% wt/vol, respectively. The samples were then concentrated for viral particles by centrifugation at 12,000 × $g$ for 2 hours at 4°C (Fig. 1). After the centrifugation step, the supernatants were discarded, and the residual viral pellets were re-suspended in 800 µL of nuclease-free water for viral RNA extraction. Using a Microlab STAR Automated Liquid Handling System (Hamilton, Reno, NV, USA), SARS-CoV-2 RNA was extracted using 200 µL of the concentrated wastewater samples using the Wastewater Large-Volume TNA Capture Kit (Promega, Madison, WI, USA) following the manufacturer's standard protocol. The 50 µL final elution was placed into a 96-well plate for RT-ddPCR and RT-qPCR assay-based detection.

## SARS-CoV-2 quantification using LDTs (RT-qPCR and RT-ddPCR methods)

The RT-qPCR LDT (Promega, Madison, WI, USA) targeted the *N* gene (N1 and N2 primers and probes developed by CDC) of SARS-CoV-2 to detect and quantify the virus. The primer and probe details, RT-qPCR cycling conditions, and performance characteristics of this assay have been previously described (15, 16). The RT-ddPCR LDT (Bio-Rad) targeted the N1 and N2 genes for viral detection and quantification as previously described (15, 16). The limit of detection (LOD) of both assays had previously been determined to be approximately 50 copies/mL.

## Specimen stability assessment

To assess specimen stability, we conducted triplicate testing of three additional clarified wastewater specimens using the FDA EUA assay after each of the four different storage conditions: (i) fresh, never frozen (baseline sample) tested within 12 hours of receipt in the laboratory (FR); (ii) 10 months of −80°C storage tested directly after thaw (1X); (iii) 10 months of −80°C storage followed by three freeze-thaw cycles (3X); and (iv) 10 months of −80°C storage followed by refrigeration (4°C) for 24 hours (1X4C).

## Data analysis

To perform the statistical analysis, we used GraphPad Prism (Version 10.1.0). One-way analysis of variance (ANOVA) with the Tukey multiple comparison test was used for the quantification of the FDA EUA assay, LDT RT-ddPCR, and LDT RT-qPCR methods (Fig. 2). Bland-Altman analysis was used to compare results obtained by the FDA EUA assay to those of the LDTs (Fig. 3 and 4). One-way ANOVA with Bonferroni's multiple comparison test was used for the quantification of sample stability (Fig. 5).

## RESULTS

### Comparison of FDA EUA assay, LDT RT-ddPCR, and LDT RT-qPCR for SARS-CoV-2 detection and quantification in wastewater samples

Looking first at a single representative WWTP, 28 samples were collected from October 2020 to February 2021 from Nashua, NH, USA. All samples were positive for detection of SARS-CoV-2 RNA using the LDTs and the FDA EUA assay (Fig. 2). Cq values from the FDA EUA assay converted to log copies per milliliter showed a mean of 2.16 log copies/mL across all Nashua samples with an upper limit of 2.53 log copies/mL and a lower limit of

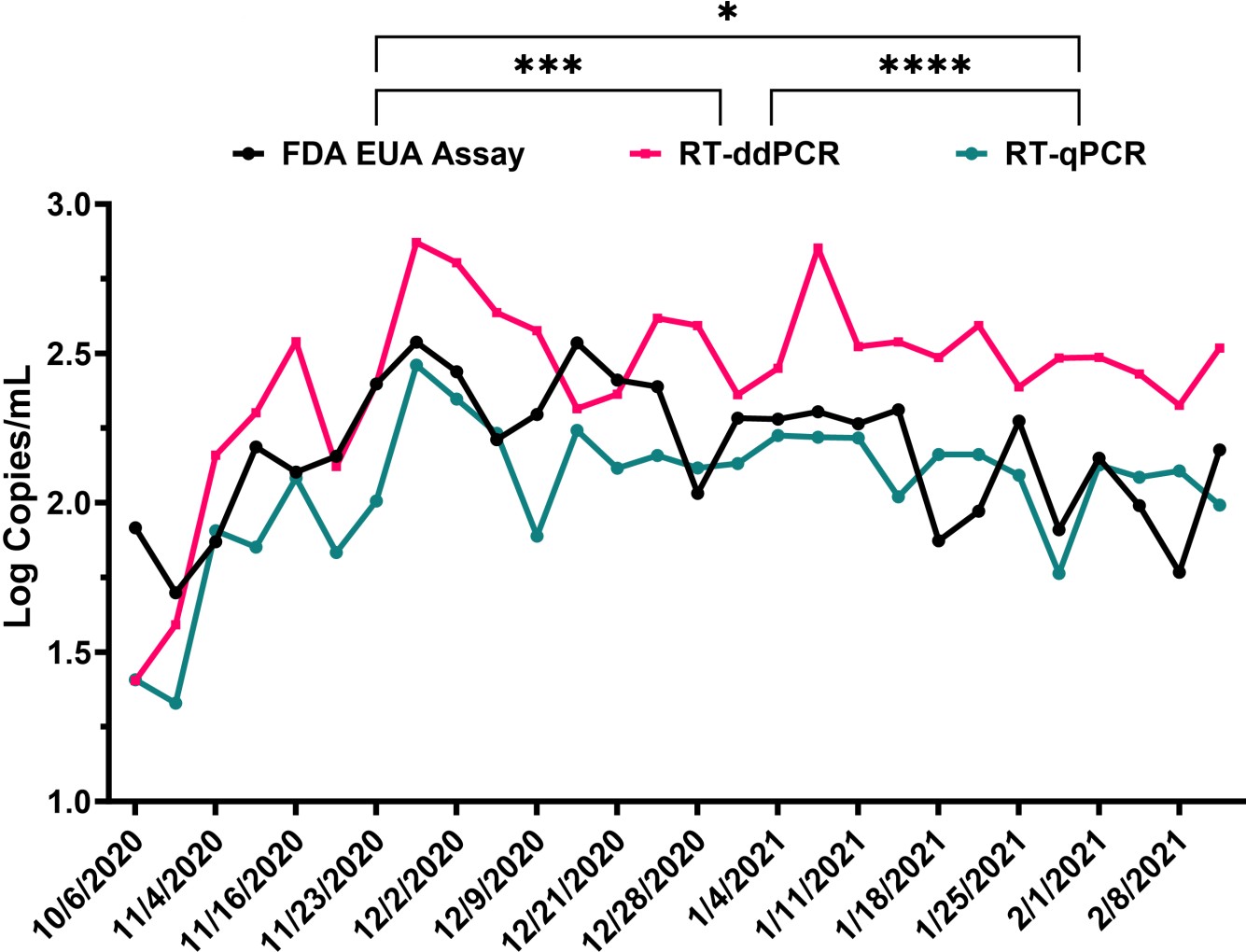

**FIG 2** Comparison of SARS-CoV-2 RNA detection between the FDA EUA assay, RT-ddPCR, and RT-qPCR. SARS-CoV-2 viral concentration (log copies per milliliter) in wastewater samples collected from the Nashua, NH WWTP over time using the FDA EUA (black), RT-ddPCR (pink), and RT-qPCR (green) methods. ns, $P > 0.05$; $*P \leq 0.05$; $***P \leq 0.001$; and $****P \leq 0.0001$.

1.69 log copies/mL. In comparison, the RT-ddPCR LDT showed overall higher values with an upper limit of 2.87 log copies/mL, a lower limit of 1.40 log copies/mL, and a mean of 2.41 log copies/mL, demonstrating 0.71 log copies/mL mean increase when compared to the FDA EUA assay ($P = 0.0001$). Similarly, the RT-qPCR LDT showed an upper limit of 2.46 log copies/mL, a lower limit of 1.32 log copies/mL, and a mean of 2.04 log copies/mL, demonstrating 0.12 log copies/mL mean decrease compared to the FDA EUA assay ($P = 0.01$).

We next evaluated the assay correlation between the FDA EUA assay and the LDTs across wastewater samples collected from multiple municipal WWTPs. We included 103 samples collected from October 2020 to February 2021. Bland-Altman analysis for the FDA EUA assay and the RT-ddPCR LDT demonstrated a correlation between the two methods with a mean difference bias of −0.42, with an upper limit of two standard deviations (SDs) of 0.40 and a lower limit SD of −1.25 (Fig. 3). Similarly, the FDA EUA assay and RT-qPCR LDT showed a correlation between methods with a mean difference of −0.13, with an upper limit of two SDs of 0.76 and a lower limit SD of −1.04 (Fig. 4). Lastly, the correlation between the RT-qPCR LDT and the RT-ddPCR LDT was confirmed, with a mean difference bias of −0.28, an upper limit of 0.24, and a lower limit of −0.81 (Fig. S3).

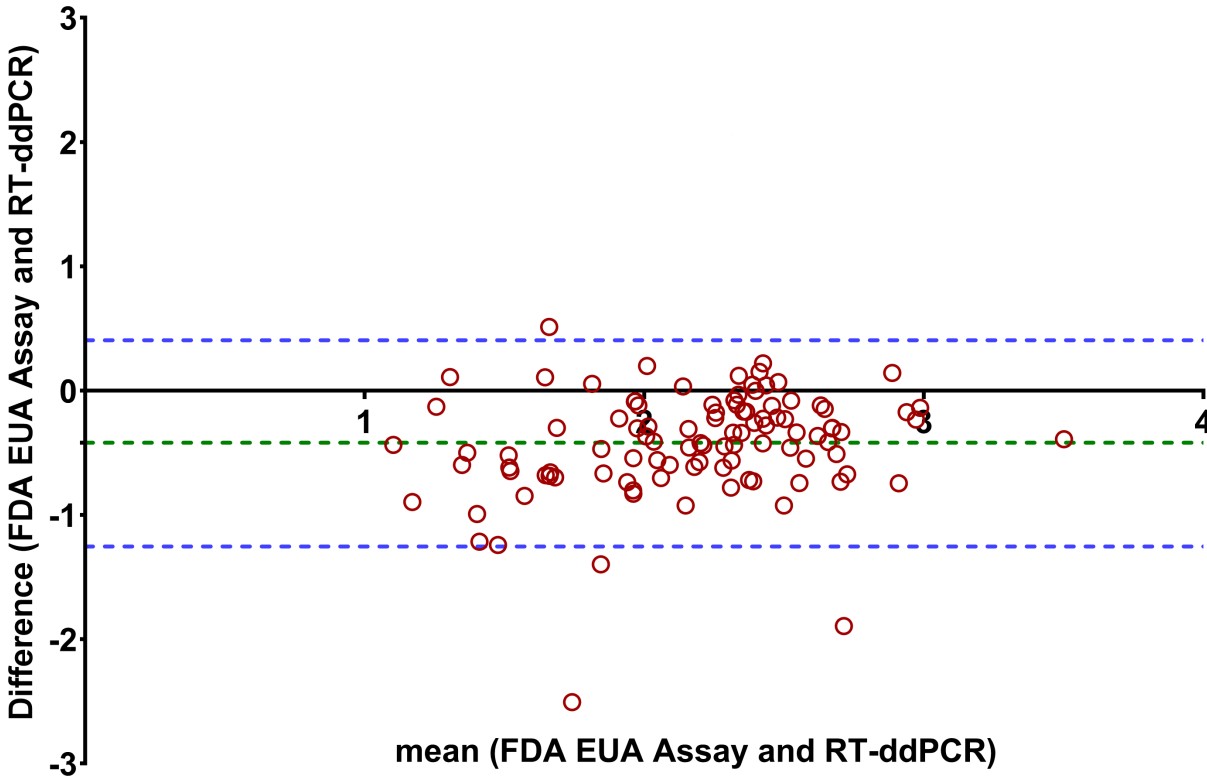

**FIG 3** A Bland-Altman plot comparing the FDA EUA and RT-ddPCR assays. The blue dotted lines represent the upper and lower limit of two standard deviations (±2 SD) of the mean, and the green dotted lines demonstrate the mean bias between the FDA EUA assay and LDT RT-ddPCR.

## Specimen stability assessment using the FDA EUA assay

The sample stability assessment demonstrated no statistically significant decline in stability in the 1X (82.1%) and 1X4C (79.4%) samples compared to the FR sample (100%). In contrast, we observed a significant decline in stability in the 3X sample (66.1%) (Fig. 5; Fig. S4). The FR wastewater aliquots from December 2020 that were tested fresh showed higher log copies per milliliter (on a mean of 2.92 log copies/mL) than the aliquots that underwent freeze-thaw cycles or prolonged refrigeration (Fig. 5; Fig. S4). The 1X samples demonstrated a mean value of 2.35 log copies/mL ($P = 0.07$), whereas the 3X samples demonstrated the lowest mean value of all conditions compared: 2.06 log copies/mL ($P = 0.001$). The 1X4C samples displayed a mean value of 2.25 log copies/mL ($P = 0.09$).

## DISCUSSION

In this proof-of-principle study, we evaluated the detection of SARS-CoV-2 RNA in wastewater samples, comparing an off-label use of an FDA EUA assay with two LDTs. Although other groups have sought to automate the process of WBS for SARS-CoV-2, this study is the first to assess the performance of off-label use of an automated FDA EUA assay for WBS (17, 18). The use of two LDTs as comparators and the inclusion of samples from multiple collection sites over a several-month timeframe are additional strengths of the study.

A strong correlation was observed among all three assays, with the FDA EUA assay showing a mean decrease of 0.42 log copies/mL and 0.13 log copies/mL compared to the RT-ddPCR and RT-qPCR LDTs, respectively (Fig. 3 and 4). Assessment of specimen stability is an important aspect of ensuring quality in SARS-CoV-2 testing in any context. The use of fresh wastewater sample demonstrated the highest RNA recovery of 2.92 log copies/mL, whereas the 3X samples showed the lowest recovery of 2.06 log copies/mL. Multiple variables can affect RNA recovery in wastewater, including duration

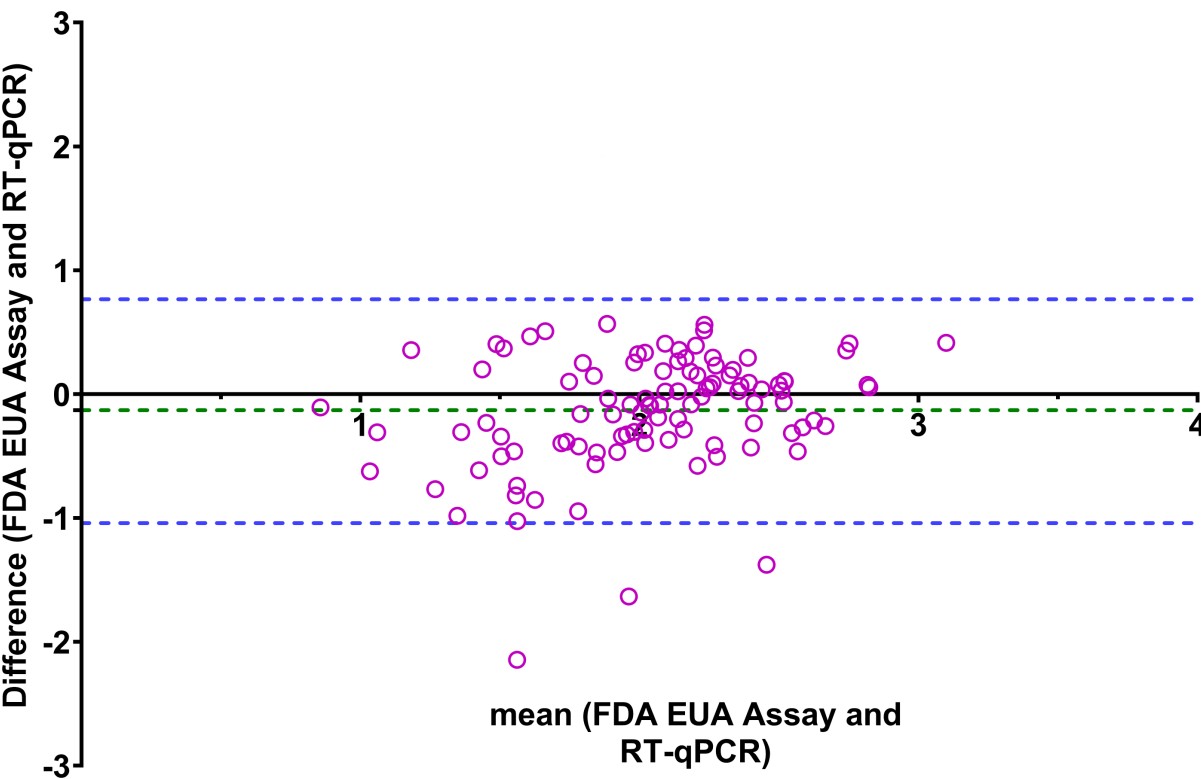

**FIG 4** A Bland-Altman plot comparing the FDA EUA and RT-qPCR assays. The blue dotted lines represent the upper and lower limit of two standard deviations (±2 SD) of the mean, and the green dotted lines demonstrate the mean bias between the FDA EUA assay and LDT RT-qPCR.

and temperature of storage (−20°C versus −80°C), whether samples are stored before or after removal of solids and/or specimen concentration, method of detection used, gene regions targeted by the assay, and presence of inhibitors. While we observed a decrease in RNA recovery of SARS-CoV-2 in clarified wastewater after multiple freeze-thaw cycles, this has not been found by all authors (19, 20). The uncertainty around optimal specimen transport and storage conditions if testing cannot be performed immediately highlights the importance of additional studies to further evaluate specimen stability in various storage conditions before adoption of the FDA EUA for WBS purposes.

Because a key purpose of WBS is to provide an early warning signal of increased pathogen transmission within a community, assays that provide a quicker total turnaround time are beneficial. The total turnaround time of the FDA EUA assay was 5 hours, which included sample processing, sample preparation, RNA extraction, PCR plate setup, and final reaction with reporting (Fig. 6). For the RT-ddPCR LDT, the turnaround time was around 8 hours and 30 minutes, and for the RT-qPCR LDT, it was 7 hours and 50 minutes.

Hands-on operator time is an important aspect to consider when selecting a test method in any laboratory. The FDA EUA assay demonstrated a significant advantage in this area over the two LDTs with only 30 minutes of operator hands-on time and minimal sample processing and preparation steps. The LDTs needed 1 hour and 55 minutes and 2 hours and 10 minutes of operator hands-on time for RT-ddPCR and RT-qPCR, respectively, with multiple sample processing and preparation steps. As the shortage of medical laboratory scientists and medical laboratory technicians continues to worsen, efficiencies in workflow become ever more important in laboratories' ability to perform testing for both clinical diagnostics and disease surveillance purposes (21).

While the workflow of the FDA EUA on the *m*2000 RealTi*m*e System is advantageous, the automated platform itself requires routine maintenance. Care must be taken with sample loading and unloading, thawing and centrifugation of reagents, cleaning, and

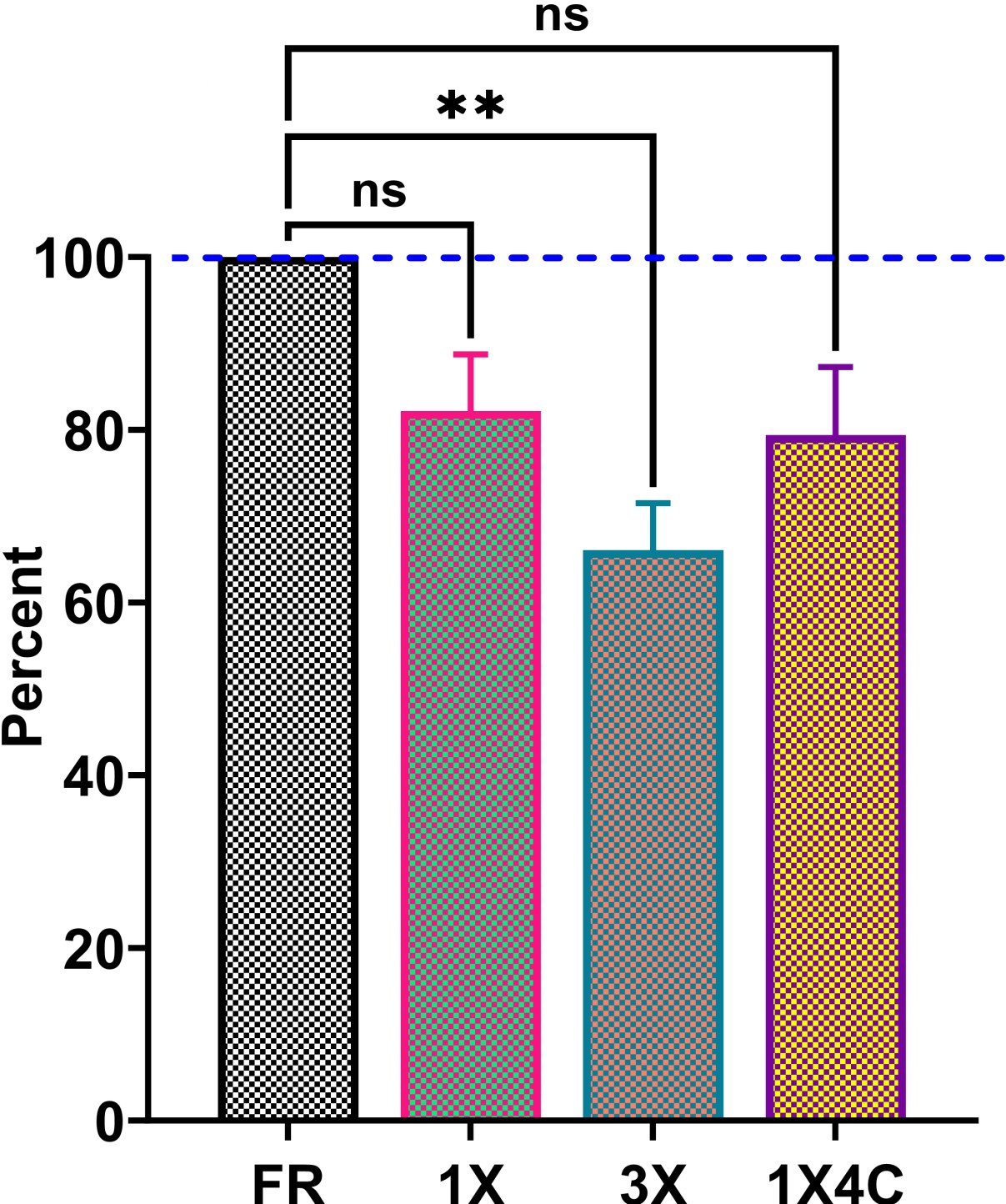

**FIG 5** Composite data for wastewater sample stability studies. Three wastewater samples were tested in triplicate using the FDA EUA assay. Normalized to the fresh wastewater samples (dashed lines at 100%). FR, fresh, never-frozen sample; 1X, 10 months of −80°C storage; 1X4C, 10 months of −80°C storage followed by refrigeration (4°C) for 24 hours; and 3X, 10 months of −80°C storage followed by three freeze-thaw cycles. ns, $P > 0.05$; **$P \leq 0.01$; ***$P \leq 0.001$; and ****$P \leq 0.0001$.

sterilization of the whole system after each run, monitoring the stability of onboard reagents, and maintenance of the system hardware (22). Such systems are therefore optimally used for testing of high specimen volumes. Additionally, even though reagents or consumables of the FDA EUA assay were affordable and convenient compared to those of the LDTs in our study (Fig. 6), the capital investment to acquire testing platforms

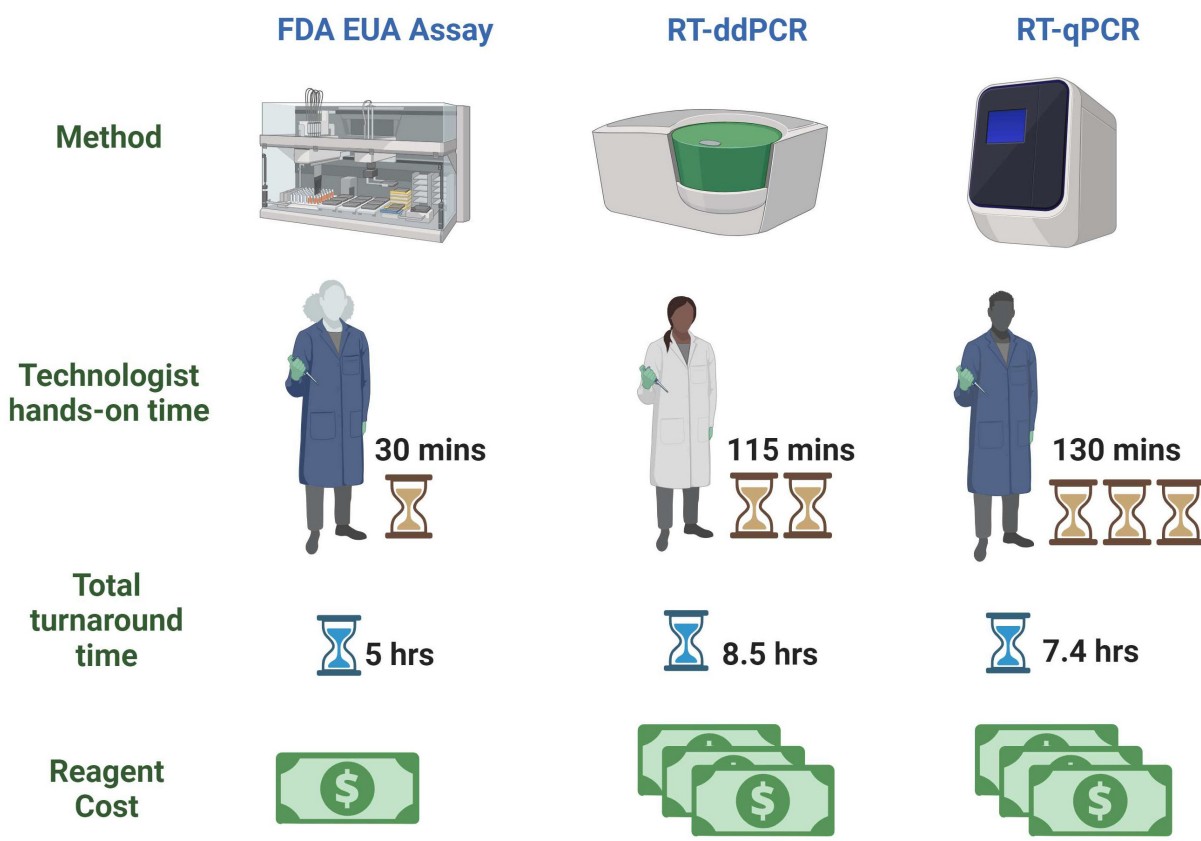

**FIG 6** Schematic representation of technologist hands-on time, turnaround time, and reagent cost between the FDA EUA assay and LDTs.

for any of the methods included in this study is significant and can vary depending on multiple factors (outright purchase versus reagent rental, overall contracting with manufacturer, etc.).

This study has several limitations. First, testing was performed during the peak of the Delta variant wave and transition to the Omicron variant wave. While the FDA EUA assay successfully detects these and subsequent variants of concern described to date as positive for SARS-CoV-2, it does not distinguish between them. Since the time the experiments in this study were performed, LDTs were developed in laboratories participating in the National Wastewater Surveillance System, which not only detect but also provide sequence information to differentiate the SARS-CoV-2 RNA present in wastewater. This enables monitoring for the emergence and spread of variants of concern (23). Because the FDA EUA assay can detect only the presence or absence of SARS-CoV-2 RNA in wastewater but cannot further characterize it, the assay is of no utility for the differentiation of emerging variants of concern. However, it can provide a convenient, early warning signal of COVID-19 spread in a community. The inclusion of regions in both the *N* and *RdRp* genes as targets in the FDA EUA assay makes it less likely that mutations in future virus variants will lead to false-negative result (22, 24, 25). Second, the FDA EUA assay is marketed as a qualitative test. The range of concentrations of SARS-CoV-2 RNA measured during the course of the study was narrow, likely due to the overall low numbers of active cases per capita in the communities monitored during the study period as well as the dilute nature of wastewater (15). Reporting of quantitative values would therefore require further validation and evaluation of a broader range of viral concentrations. Such a validation would include determination of the assay's performance characteristics including linear range and lower LOD. The manufacturer of the FDA EUA assay claims an LOD of 50 copies/mL (1.7 log copies/mL)

for clinical samples (for example, nasopharyngeal samples collected in viral transport medium), and although a similar LOD might be expected with wastewater, the use of an alternative matrix or specimen type would require a formal LOD determination to ensure low-level concentrations of virus can be reproducibly detected. Lastly, robust precision (reproducibility) studies would need to be completed for full validation to understand the variability inherent in the test method.

In conclusion, the FDA EUA assay used with the *m*2000 RealTi*me* System performed comparably to two LDTs for qualitative detection of SARS-CoV-2 RNA in wastewater samples. The operator hands-on time and overall turnaround time of the FDA EUA assay offer efficiency in comparison to the RT-qPCR and RT-ddPCR assays, the former being a critical benefit in this era of widespread laboratory staffing challenges. Our study provides proof of concept that an FDA EUA assay may provide a streamlined labor- and time-saving method of WBS for SARS-CoV-2. Although the COVID-19 public health emergency has now ended, lessons about the application of automated technology developed for human diagnostics to WBS in future pandemics could prove useful.

## ACKNOWLEDGMENTS

We thank the wastewater treatment facility employees who collected samples for each of the collection dates from our nine study sites. We also thank the couriers of Green Mountain Messenger for their handling and timely delivery of wastewater specimens. We appreciate our University of New Hampshire collaborators, Paula Mouser and Fabrizio Colosimo, for their feedback and support.

## AUTHOR AFFILIATIONS

[1]Department of Pathology and Laboratory Medicine, Dartmouth Health, Lebanon, New Hampshire, USA
[2]The Broad Institute at MIT and Harvard, Cambridge, Massachusetts, USA
[3]Hubbard Lab Consulting, Coraopolis, Pennsylvania, USA

## AUTHOR ORCIDs

Joel A. Lefferts  http://orcid.org/0000-0002-5083-7867
Isabella W. Martin  http://orcid.org/0000-0001-7974-9767

## AUTHOR CONTRIBUTIONS

Shivaprasad H. Sathyanarayana, Data curation, Formal analysis, Investigation, Writing – original draft, Writing – review and editing | Ashlee A. Robins, Conceptualization, Data curation, Investigation, Writing – review and editing | Diana M. Toledo, Conceptualization, Data curation, Investigation, Writing – review and editing | Torrey L. Gallagher, Conceptualization, Data curation, Formal analysis, Project administration, Resources, Supervision, Writing – review and editing | Gregory J. Tsongalis, Conceptualization, Project administration, Resources, Writing – review and editing | Jacqueline A. Hubbard, Conceptualization, Data curation, Formal analysis, Funding acquisition, Investigation, Project administration, Validation, Visualization, Writing – review and editing | Joel A. Lefferts, Conceptualization, Data curation, Formal analysis, Methodology, Supervision, Visualization, Writing – review and editing | Isabella W. Martin, Conceptualization, Data curation, Formal analysis, Funding acquisition, Methodology, Project administration, Supervision, Writing – review and editing

## ADDITIONAL FILES

The following material is available online.

## Supplemental Material

**Supplemental material (Spectrum02490-24-s0001.docx).** Supplemental methods, tables, and figures.

## Open Peer Review

**PEER REVIEW HISTORY (review-history.pdf).** An accounting of the reviewer comments and feedback.

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
