## [Reviewer comments · Microbiology Spectrum]

Microbiology Spectrum

Simplifying SARS-CoV-2 Wastewater-based Surveillance Using an Automated FDA EUA Assay

Shivaprasad Sathyanarayana, Ashlee Robbins, Diana Toledo, Torrey Gallagher, Gregory Tsongalis, Jacqueline Hubbard, Joel Lefferts, and Isabella Martin

Corresponding Author(s): Isabella Martin, Dartmouth-Hitchcock Medical Center

Review Timeline:

Submission Date:	October 17, 2024
Editorial Decision:	December 3, 2024
Revision Received:	February 6, 2025
Accepted:	February 8, 2025

Editor: Katharina Kujala

Reviewer(s): Disclosure of reviewer identity is with reference to reviewer comments included in decision letter(s). The following individuals involved in review of your submission have agreed to reveal their identity: Megan Elizabeth Jane Lott (Reviewer #1)

Transaction Report:

DOI: <https://doi.org/10.1128/spectrum.02490-24>

Re: Spectrum02490-24 (Simplifying SARS-CoV-2 Wastewater-based Surveillance Using an Automated FDA EUA Assay)

Dear Dr. Isabella W. Martin:

Thank you for the privilege of reviewing your work. Below you will find my comments, instructions from the Spectrum editorial office, and the reviewer comments.

Revision Guidelines

Sincerely,
Katharina Kujala
Editor
Microbiology Spectrum

Reviewer #2 (Comments for the Author):

The manuscript compares three different workflows to detect Sars-CoV-2 from wastewater. The manuscript is presented precisely.
Comments;

1. The author compared the workflows and time required for each process. Adding a comparison of the associated costs for each workflow would further enhance the readers' understanding of the operational feasibility of the processes.
2. The author compared the log copies/ml for the test samples. Kindly include the repeatability and reproducibility of the chosen methods to better understand the variation introduced by the method.
3. Comprehensive validation is essential, irrespective of the output format, kindly include the spiked sample data and linearity graph.
4. Line 135; Calibration using spiked control, kindly include the data as supplementary.
5. The author explored the impact of storage and freeze-thaw cycles. This section could be expanded to emphasize the significance of freeze-thaw cycles and storage conditions.

In this paper, the authors evaluate the off-label use of an automated sample preparation and RT-dPCR platform (from Abbott Molecular) for detection and quantification of SARS-CoV-2 RNA from wastewater. The platform has received an Emergency Use Authorization from the FDA for confirmation of SARS-CoV-2 detection from clinical samples with suspected cases of COVID-19. After adapting the workflow for wastewater, the authors compare the method to those conventionally used for wastewater monitoring of SARS-CoV-2 – sample concentration, RNA isolation, and molecular analysis by RT-qPCR and RT-ddPCR. The results, compared using a Bland-Altman analysis, were comparable between the denovo workflow and the conventional workflows.

This paper is well-crafted. The study design is well-constructed and the results are well-articulated. Figure 6 is helpful in underscoring the added value of this workflow (shorter workflow, with less hands-on time). This paper is likely to be well-accepted by the readers of mSpectrum.

The authors may consider the following suggestions:

- Including a Blank-Altman plot comparing the agreement between the RT-dPCR and RT-qPCR platforms.
- Figure 5 – The goal of this figure is to compare the concentration of SARS-CoV-2 in the wastewater samples following three different storage conditions (fresh, one freeze-thaws, three freeze-thaws, and one freeze-thaw with a hold at 4C. It appears that Figure 5A and Figure 5B represent the same data, just in two different ways. It may be more appropriate to move Figure 5A to the supplemental, and to generate a new figure 5 that shows the average concentration per storage treatment. Then, you can add the results of the paired statistical test.
- Figure 5B – what does the dashed line represent?

The manuscript compares three different workflows to detect Sars-CoV-2 from wastewater. The manuscript is presented precisely.

Comments;

1. The author compared the workflows and time required for each process. Adding a comparison of the associated costs for each workflow would further enhance the readers' understanding of the operational feasibility of the processes.
2. The author compared the log copies/ml for the test samples. Kindly include the repeatability and reproducibility of the chosen methods to better understand the variation introduced by the method.
3. Comprehensive validation is essential, irrespective of the output format, kindly include the spiked sample data and linearity graph.
4. Line 135; Calibration using spiked control, kindly include the data as supplementary.
5. The author explored the impact of storage and freeze-thaw cycles. This section could be expanded to emphasize the significance of freeze-thaw cycles and storage conditions.

Dear Reviewers:

Thank you for the thoughtful and helpful feedback. We appreciate the opportunity to revise and resubmit our manuscript. We have addressed all the issues raised by the reviewers. Please find our responses below in blue text.

Reviewer #1:

In this paper, the authors evaluate the off-label use of an automated sample preparation and RT-dPCR platform (from Abbott Molecular) for detection and quantification of SARS-CoV-2 RNA from wastewater. The platform has received an Emergency Use Authorization from the FDA for confirmation of SARS-CoV-2 detection from clinical samples with suspected cases of COVID-19. After adapting the workflow for wastewater, the authors compare the method to those conventionally used for wastewater monitoring of SARS-CoV-2 – sample concentration, RNA isolation, and molecular analysis by RT-qPCR and RT-ddPCR. The results, compared using a BlandAltman analysis, were comparable between the denovo workflow and the conventional workflows.

This paper is well-crafted. The study design is well-constructed and the results are well-articulated. Figure 6 is helpful in underscoring the added value of this workflow (shorter workflow, with less hands-on time). This paper is likely to be well-accepted by the readers of mSpectrum.

Thank you for this thoughtful and positive summary of our work.

The authors may consider the following suggestions:

1. Including a Bland-Altman plot comparing the agreement between the RT-dPCR and RT-qPCR platforms.

Thank you for this great suggestion. We have made a Bland-Altman plot comparing RT-ddPCR and RT-qPCR platforms and included it in the Supplemental Materials since we had already reached our maximum number of Figures for the manuscript. We have also added the following text to the Results section (Lines 199-201):

“Lastly, the correlation between the RT-qPCR LDT and the RT-ddPCR LDT was confirmed, with a mean difference bias of -0.28, an upper limit of 0.24, and a lower limit of -0.81 (**Supplemental Materials**).”

2. Figure 5 – The goal of this figure is to compare the concentration of SARS-CoV-2 in the wastewater samples following three different storage conditions (fresh, one freeze-thaws, three freeze-thaws, and one freeze-thaw with a hold at 4C. It appears that Figure 5A and Figure 5B represent the same data, just in two different ways. It may be more appropriate to move Figure 5A to the supplemental, and to generate a new figure 5 that

shows the average concentration per storage treatment. Then, you can add the results of the paired statistical test.

Thank you for this helpful suggestion. As recommended, we have moved Figure 5A to the Supplemental Material and now have a standalone Figure 5 (previously Figure 5B) showing percent decrease from baseline value after the various storage conditions. If we had averaged the concentration values for Samples 1, 2 and 3 together at each storage condition as suggested, the error bars would be prohibitively large because the baseline values for the three samples were quite different. We therefore chose to keep the original format of this figure.

3. Figure 5B – what does the dashed line represent?

Thank you for noticing the erroneous placement of the dashed line in what is now Figure 5. It was intended to be a visual marker of the 100% level, but was not in the correct place. We have corrected the placement of the line and modified the figure legend to make clear what it represents.

Reviewer #2:

The manuscript compares three different workflows to detect Sars-CoV-2 from wastewater. The manuscript is presented precisely.

Comments:

1. The author compared the workflows and time required for each process. Adding a comparison of the associated costs for each workflow would further enhance the readers' understanding of the operational feasibility of the processes.

We agree that knowing the relative cost of different assays is helpful information for readers. Cost of commercial reagents can be highly variable depending on volumes tested and overall contracting between an institution and commercial manufacturer. While we cannot share specific numbers, we have included relative cost of reagents in Figure 7 and added the following to our discussion (Lines 259-263):

“Additionally, even though reagents or consumables of the FDA EUA assay were affordable and convenient compared to those of the LDTs in our study (**Figure 6**), the capital investment to acquire testing platforms for any of the methods included in this study is significant and can vary depending on multiple factors (outright purchase versus reagent rental, overall contracting with manufacturer, etc.).”

2. The author compared the log copies/ml for the test samples. Kindly include the repeatability and reproducibility of the chosen methods to better understand the variation introduced by the method.

We agree that repeatability/reproducibility is an extremely important performance characteristic to evaluate in a method validation and one would want to perform a thorough evaluation before using an assay for clinical testing or wastewater surveillance. Because our study was not a full validation study but rather a proof-of-principal study to explore feasibility, we did not perform extensive reproducibility studies aside from triplicate testing of the 3 samples used for stability studies. This limitation has been added to the Discussion section (Lines 288-289):

“Lastly, robust precision (reproducibility) studies would need to be completed for full validation to understand the variability inherent in the test method.”

3. Comprehensive validation is essential, irrespective of the output format, kindly include the spiked sample data and linearity graph.

Thank you for making this recommendation. We agree that comprehensive validation is necessary before testing is put into place for public health or clinical purposes. Our study was not one of comprehensive validation, but instead a “proof-of-principal” study to explore feasibility of using an automated FDA-cleared platform for this testing. We note this limitation in our Discussion section (Lines 266-271). We have provided more detail about the spiked sample study and linearity graph in the newly-created Supplemental Material. Additionally, we have edited the text in the Materials and Methods section as follows (Lines 135-137):

“We employed SeraCare quantitative calibration material and positive pooled sample dilutions to create a dilution series and calibration curve that was tested against reference material spiked into negative clarified wastewater samples (Supplemental Materials)”

4.Line 135; Calibration using spiked control, kindly include the data as supplementary.

Thank you for this comment. We have included more detail about the calibration study in the supplementary file and edited the text in the Materials and Methods section as described above.

5. The author explored the impact of storage and freeze-thaw cycles. This section could be expanded to emphasize the significance of freeze-thaw cycles and storage conditions.

This is an excellent point. We have edited and expanded the discussion to read as follows and added two new references (Lines 227-235):

“Multiple variables can affect RNA recovery in wastewater, including duration and temperature of storage (-20°C versus -80°C), whether samples are stored before or after removal of solids and/or specimen concentration, method of detection used, gene regions targeted by the assay and presence of inhibitors. While we observed a decrease in RNA recovery of SARS-CoV-2 in clarified wastewater after multiple freeze-

thaw cycles, this has not been found by all authors.^{19,20} The uncertainty around optimal specimen transport and storage conditions if testing cannot be performed immediately highlights the importance of additional studies to further evaluate specimen stability in various storage conditions before adoption of the FDA EUA for WBS purposes.”

Re: Spectrum02490-24R1 (Simplifying SARS-CoV-2 Wastewater-based Surveillance Using an Automated FDA EUA Assay)

Dear Dr. Isabella W. Martin:

Thank you for addressing all reviewer comments in your resubmission. I am pleased to inform you that your manuscript has been accepted, and I am forwarding it to the ASM production staff for publication. Your paper will first be checked to make sure all elements meet the technical requirements. ASM staff will contact you if anything needs to be revised before copyediting and production can begin. Otherwise, you will be notified when your proofs are ready to be viewed.

Sincerely,
Katharina Kujala
Editor
Microbiology Spectrum